# GFI1-Dependent Repression of *SGPP1* Increases Multiple Myeloma Cell Survival

**DOI:** 10.3390/cancers14030772

**Published:** 2022-02-02

**Authors:** Daniela N. Petrusca, Patrick L. Mulcrone, David A. Macar, Ryan T. Bishop, Evgeny Berdyshev, Attaya Suvannasankha, Judith L. Anderson, Quanhong Sun, Philip E. Auron, Deborah L. Galson, G. David Roodman

**Affiliations:** 1Department of Medicine, Hematology/Oncology Division, Indiana University School of Medicine, 980 Walnut St., Indianapolis, IN 46202, USA; pamulcro@iu.edu (P.L.M.); asuvanna@iu.edu (A.S.); andejul@iu.edu (J.L.A.); groodman@iu.edu (G.D.R.); 2Department of Biological Sciences, Duquesne University, 600 Forbes Ave., Pittsburgh, PA 15219, USA; macard@duq.edu (D.A.M.); auronp@duq.edu (P.E.A.); 3Department of Tumor Biology, H. Lee Moffitt Cancer Research Center and Institute, 12902 USF Magnolia Drive, Tampa, FL 33612, USA; Ryan.Bishop@moffitt.org; 4Department of Medicine, National Jewish Health, 1400 Jackson Street, Denver, CO 80206, USA; BerdyshevE@NJHealth.org; 5Richard L. Rodebush Veterans Affairs Medical Center, 1481 W 10th St., Indianapolis, IN 46202, USA; 6Department of Medicine, Division of Hematology/Oncology, McGowan Institute for Regenerative Medicine, University of Pittsburgh, UPMC Hillman Cancer Center Research Pavilion, 5117 Centre Ave, Pittsburgh, PA 15213, USA; sunq2@upmc.edu (Q.S.); galson@pitt.edu (D.L.G.)

**Keywords:** GFI1, multiple myeloma, SGPP1, S1P, c-Myc, PP2A

## Abstract

**Simple Summary:**

New therapies have greatly improved the progression-free and overall survival for patients with “standard risk” multiple myeloma (MM). However, patients with “high risk” MM, in particular patients whose MM cells harbor non-functional p53, have very short survival times because of the early relapse and rapid development of highly therapy-resistant MM. In this report, we identify a novel mechanism responsible for Growth Factor Independence-1 (GFI1) regulation of the growth and survival of MM cells through its modulation of sphingolipid metabolism, regardless of their p53 status. We identify the Sphingosine-1-Phosphate Phosphatase (SGPP1) gene as a novel direct target of GFI1 transcriptional repression in MM cells, thus increasing intracellular sphingosine-1-phosphate levels, which stabilizes c-Myc. Our results support GFI1 as an attractive therapeutic target for all types of MM, including the “high risk” patient population with non-functional p53, as well as a possible therapeutic approach for other types of cancers expressing high levels of c-Myc.

**Abstract:**

Multiple myeloma (MM) remains incurable for most patients due to the emergence of drug resistant clones. Here we report a p53-independent mechanism responsible for Growth Factor Independence-1 (GFI1) support of MM cell survival by its modulation of sphingolipid metabolism to increase the sphingosine-1-phosphate (S1P) level regardless of the p53 status. We found that expression of enzymes that control S1P biosynthesis, *SphK1*, dephosphorylation, and *SGPP1* were differentially correlated with GFI1 levels in MM cells. We detected GFI1 occupancy on the *SGGP1* gene in MM cells in a predicted enhancer region at the 5’ end of intron 1, which correlated with decreased *SGGP1* expression and increased S1P levels in GFI1 overexpressing cells, regardless of their p53 status. The high S1P:Ceramide intracellular ratio in MM cells protected c-Myc protein stability in a PP2A-dependent manner. The decreased MM viability by SphK1 inhibition was dependent on the induction of autophagy in both p53WT and p53mut MM. An autophagic blockade prevented GFI1 support for viability only in p53mut MM, demonstrating that GFI1 increases MM cell survival via both p53WT inhibition and upregulation of S1P independently. Therefore, GFI1 may be a key therapeutic target for all types of MM that may significantly benefit patients that are highly resistant to current therapies.

## 1. Introduction

Multiple myeloma (MM) is the second most common hematological malignancy [1] and the most frequent cancer involving bone. MM remains incurable for the vast majority of patients [1,2,3,4] due to the emergence of drug resistant clones and mutations inducing drug resistant relapses [5]. Thus, identifying new therapeutic targets in MM patients that can overcome drug resistance is critically needed.

Growth Factor Independence 1 (GFI1) is a Zn-finger DNA binding protein with an N-terminal SNAG domain that recruits repressive chromatin modifiers, such as lysine demethylase 1 (LSD1, KDM1A), to target genes to exert its role as a transcriptional repressor [6]. GFI1 is known for its critical roles in hematopoietic stem cells as well as in the development, specification, and function of multiple hematopoietic lineages by repressing key target genes [7,8]. High levels of GFI1 in T and B cells abolish G1 cell cycle arrest and apoptosis induced by growth factor withdrawal [7,9]. Besides regulating normal hematopoiesis, high levels of GFI1 have been associated with supporting the viability of both types of acute lymphoblastic leukemia (ALL) in the presence of initiating oncogenes. At least part of the mechanism by which GFI1 supports the survival of T-ALL is by prevention of p53-induced apoptosis via GFI1 interaction with the p53 C-terminus leading to the recruitment of LSD1 to demethylate p53, which decreases the acetylation of p53—thus blocking p53 from activating pro-apoptotic target genes [10,11]. 

We recently reported that GFI1 mediates MM cell growth and viability, enhances MM cell resistance to bortezomib-induced cell death in vitro, and increases MM cell growth and osteoclastogenesis in vivo [12]. We found a p53-dependent mechanism responsible for GFI1 repression of MM cell death, but also demonstrated that GFI1 is essential for MM cell survival regardless of their p53 status [12]. Therefore, GFI1 must also act through a p53-independent mechanism in MM cells. During our earlier studies, we showed that GFI1 levels were increased by the administration of MM supportive signals from the microenvironment, such as extracellular S1P. However, the effects of GFI1 on intracellular sphingolipid (SL) metabolism have not been investigated. 

Sphingolipid metabolism (Appendix A) is a fundamental pathway of tumor cell survival. SLs are bioactive lipids critically involved in the cellular response to stress stimuli and act as important regulators of cell fate [13]. Ceramide (Cer) and sphingosine (SPH) are proapoptotic SLs, while sphingosine-1 phosphate (S1P) is a pleiotropic lipid mediator that regulates all major processes involved in cancer progression, including cell survival, migration, the recruitment of immune cells, and angiogenesis [14]. Abnormalities in the Cer:SPH:S1P ratio have been implicated in inflammation, autoimmune diseases, and cancers [15,16,17,18]. S1P is the metabolic product of Sphingosine kinases (SphKs). SphK1 expression is increased in numerous cancers [19,20] and has been implicated in the suppression of apoptosis in multiple cell types [21,22,23,24]. S1P phosphatase 1 (SGPP1) is responsible for degrading S1P via the salvage and recycling of SPH into long-chain ceramides, whereas spinghosine-1-phsophate lyase 1 (SGPL1) cleaves S1P into a fatty aldehyde and phosphoethanolamine [25,26].

*c-Myc* gene expression plays a key role in MM progression. It is frequently upregulated in later stages of MM and is associated with an aggressive form of MM [27]. A several-fold increase in *c-Myc* RNA expression occurs with progression of pre-malignant monoclonal gammopathy of undetermined significance (MGUS) to MM, and this key role for c-Myc in MM is strongly supported by the Vk*MYC mouse model [28]. Targeted loss of function studies suggested that Myc is required for both the proliferation and survival of MM cell lines and cultured primary MM cells and support the concept that Myc plays a key role in MM cell survival [29]. c-Myc is a master transcription factor mainly involved in the control of proliferation and differentiation, metabolism, and cell growth [30,31]. Although, in normal cells c-Myc expression is tightly controlled, aberrant c-Myc activity is frequently found in hematopoietic neoplasia [32]. Post-transcriptional regulation plays an essential role in c-Myc’s stability and function. Phosphorylation of c-Myc at Serine 62 (S62) initiated by the ERK cascade or cyclin-dependent kinases (CDKs) [33] enhances c-Myc’s stability and its DNA binding and target gene regulation. Myc (S62) can be further phosphorylated by glycogen synthase kinase 3 (GSK3)—mediated at Threonine 58 (T58) [34]. Dual phosphorylated c-Myc (S62 and T58) becomes the target of protein phosphatase 2A (PP2A), which dephosphorylates the stabilizing S62 residue, leading to c-Myc degradation [35]. Thus, PP2A activity has a fundamental role in maintaining the balance between normal and aberrant c-Myc activity.

Here, we report that the GFI1-dependent and p53-independent modulation of sphingolipid metabolism promotes MM cell survival, as well as the enhancement of MM cell growth and drug resistance induced by the BM microenvironment. We found that GFI1 acts as a direct repressor of *SGPP1* gene transcription in MM cells, which increased the intracellular S1P levels regardless of the MM cell’s p53 status. The resultant imbalance in the S1P:Cer ratio inactivates PP2A and stabilizes c-Myc expression in MM cells to increase their survival and drug resistance.

## 2. Materials and Methods

### 2.1. Human Samples

Primary CD138+ and BMSC (bone marrow stromal cells) cell purification: Bone marrow (BM) aspirates were collected in heparin from three healthy subjects and ten patients with plasma cell disorders (Appendix A). Non-adherent marrow mononuclear cells were collected, and highly purified MM cells (>90%) were isolated by magnetic cell fractionation with anti-CD138 MicroBeads (Miltenyi Biotec Inc., San Diego, CA, USA) [36], as previously described [37,38]. The remaining adherent cells were cultured for 21 days with media changes every 4 days to obtain the BMSCs that were used at passages 2 and 3. 

### 2.2. Cell Culture

Myeloma cell lines were purchased from ATCC (Manassas, VA, USA) (MM.1S; H929, OPM-2, RPMI-8226) or were generously provided by Drs. Louis Stancato (U266), Kenneth Anderson (KMS-11), Arun Wiita (AMO-1), and Nicola Giuliani (JJN3). The SAKA-T bone marrow stromal cell line was previously generated by our group [39]. The MM cells were maintained in RPMI-1640 medium supplemented with 10% FBS and 1% Pen/Strep at 37 °C with 5% CO_2_. The MLO-A5 murine osteocyte-like cells (provided by Dr. Jesús Delgado-Calle) were cultured in dishes coated with Collagen type 1α (MP Biomedicals #160084) and maintained in phenol red-free α-MEM supplemented with 5% BCS and 5% FBS at 37 °C with 5% CO_2_. The pre-osteoblast murine cell line MC3T3-E1 subclone-4 (MC4) was generously provided by Dr. Guozhi Xiao [40,41] and maintained in ascorbic-acid free α-MEM supplemented with 10% FCS and 1% Pen/Strep.

### 2.3. Plasmid and Lentiviral Constructs

The MM1.S, OPM-2, KMS-11, and RPMI-8226 cells, stably overexpressing GFI1 (GFI1 o/e), were obtained by transduction with a puc2CL6IEGwo-GFP lentivirus encoding for GFI1 or through an empty vector (EV) after selection by GFP flow cytometry using a BD FACSAria II sorter (BD Bioscience, San Jose, CA, USA), as previously described [12], and transient KD GFI1 was obtained in H929 cells using shRNA for GFI1, as previously described [12]. Stable MM.1S expressing secretable Guassia luciferase (gLuc) were obtained by transfection with a pCL6-IEG/gLuc vector followed by 48 h of puromycin selection. Stable Dox-ishGFI1 (MM.1S and OPM-2) and SPNS2 (MM.1S) cells were generated by using a SMARTvectorTM Inducible Lentiviral shRNA vector (Dharmacon; Lafayette, CO, USA) utilizing the Tet-On 3G bipartite induction system. The MM cells were transduced with either SMARTvector Human Inducible GFI1 shRNA; hEF1α; TurboGFP or SMARTvector Human Inducible SPNS2 shRNA; hEF1α; TurboGFP or SMARTvector Inducible Non-targeting Control; and hEF1α TurboGFP per the manufacturer’s protocol followed by 48 h of puromycin selection. To induce the KD, stably transduced MM cells were treated with doxycycline (1 μg/mL 3 days) that permits the TRE3G promoter to be bound and activated by the constitutively expressed Tet-On 3G transactivator protein, which is also encoded within the Inducible shRNA vector. Together, the Tet-On 3G protein and TRE3G promoter permit tight regulation of the shRNA expression and potent induction. 

### 2.4. Cell Viability Assays

Human MM cell lines were incubated in 96-well plates in RPMI-1640 media with 2% FCS and varying concentrations of SK1-I (Enzo Life Sciences, Farmingdale, NY, USA) for 24 and 48 h. Cell viability was quantified by alamarBlue^®^ Cell Viability assays (Thermos Fisher Scientific, Waltham, MA, USA), per the manufacturer’s protocol using a BioTek Synergy H1 automatic plate reader (Winooski, VT, USA). 

### 2.5. PP2A Activity Assay

Cells were scraped from the culture plates in phosphatase extraction buffer containing 20 mM imidazole-HCl, 2 mM EDTA, 2 mM EGTA, pH 7.0, and with 10 µg/mL each of aprotinin, leupeptin, antipain, soybean trypsin inhibitor, 1 mM benzamidine, and 1 mM PMSF, and then were sonicated for 10 s and centrifuged at 2000× *g* for 5 min. The supernatants were used for phosphatase activity assays per the manufacturer’s protocol (Sigma-Millipore, Saint Louise, MO, USA). In brief, 500 µg of lysate were immunoprecipitated with anti-PP2A, C subunit (clone 1D6) and the protein immunocomplexes were collected with a Protein A agarose slurry. The PP2A activity was assessed by dephosphorylation of a phosphor-peptide substrate and measured by a Malachite Green phosphate detection solution at 650 nm with a BioTek Synergy H1 automatic plate reader (Winooski, VT, USA).

### 2.6. Immunoprecipitation and Western Blotting

Cell pellets were lysed with IP lysis buffer containing 50 mM HEPES (pH7.5), 150 mM NaCl, 1 mM EGTA, 1.5 mM MgCl_2_, 1% Triton X-100, and 10% glycerol, which were supplemented with a proteinase inhibitor cocktail (Sigma, P8340). Lysates (500 μg) incubated with anti-PP2A (Millipore, 05-421) or control IgG antibodies (Invitrogen, Invitrogen, Eugene, OR, USA, MA5-14447) were stored overnight at 4° C. The protein/antibody complexes were precipitated by protein A/G magnetic beads (Invitrogen) for 2 h. The denatured protein complexes were separated by 10% SDS-PAGE gel electrophoresis and transferred to a PVDF membrane (Millipore, Temecula, CA, USA). For regular western blotting, whole cell protein lysates were extracted in a RIPA lysis buffer (sc-24948; SCB) and were supplemented with a protease inhibitors cocktail (Sigma-Aldrich, Saint Louis, MO, USA). Equal amounts of proteins, as determined by a bicinchoninic acid assay protein analysis (Pierce, Rockford, IL), were separated on 10%, 12%, or Any kD™ SDS-PAGE gels (Bio-Rad Laboratories, Hecules, CA, USA) and transferred onto PVDF membranes. For the detection of immune complexes, the membranes were incubated with various primary antibodies directed towards: GFI1 (Santa Cruz, sc-8558); P(Ser-225)-SphK1 (ECM Biosciences, SP1641); c-Myc (CST, 13987); P(S62) c-Myc (Abcam, ab185656); P-PP2A (Santa Cruz, sc-271903); I2PP2A (Santa Cruz, sc-133138); P-ERK1/2 (CST, 9101); ERK1/2 (CST, 4696) caspase 3 (CST, 9662), cleaved caspase 3 (CST, 9661); LC3B (Sigma, L7543); p62 (Abcam, ab155686); β-actin (Sigma, A5441) and VCP (Abcam, ab11433). Band signals following the incubation with specific HRP-linked secondary antibodies were detected using the enhanced chemiluminescence kit (Thermo Scientific, Rockford, IL, USA). The immune complexes were quantified by densitometry using ImageJ software after normalization to specific loading controls.

### 2.7. Immunofluorescence

Air dried cytospins containing MM cells were fixed in 10% paraformaldehyde for 10 min followed by a treatment with cold methanol at −20 °C for 30 min and three PBS washes. Cytospins were then blocked with 5% BSA containing 0.2% Triton-100 for 1 h at room temperature. Slides were incubated with rabbit anti-human SGPP1 antibody (Sigma-Aldrich, Saint Louise, MO, USA) in PBST containing 2% BSA overnight at 4 °C in a humid container. The PBS washed slides were then incubated with Alexa Fluor 568 donkey anti-rabbit IgG (H + L), Invitrogen, Eugene, OR, USA) for 1 h. Slowfade Gold antifade (Invitrogen, Eugene, OR) containing DAPI (4, 6-diamidino-2-phenylindole) was used to mount the slides. The images were taken on an Olympus IX73 (Waltham, MA, USA) microscope.

### 2.8. 3D Bone Marrow Microenvironment

For the 3D co-cultures of bone cells (MC4 pre-osteoblasts or MLOA5 osteocytes) and MM cells (MM.1S-gLUC), an equal parts mixture of basement membrane extract (BME, Culturex Trevigen #3433-005) and cells in 10% FBS α-MEM were added to 96-well plates and monitored for up to 72 h. A 1:5 ratio of bone cells to MM was used for the co-culture groups [42,43]. BME was thawed overnight at 4 °C. Cells of interest were counted and pelleted via centrifugation. The cell pellets were resuspended in 50 μL of 10% FBS α-MEM, then 50 μL of thawed BME was added to the cells. The 100 μL cells/BME mixture was then added to wells of a 96-well plate and incubated at 37 °C for 30 min. After incubation, 125 μL of 10% FBS α-MEM (bone cells alone), 10% FBS RPMI (MM cell alone), or a 1:1 mixture of 10% FBS α-MEM and 10% FBS RPMI (co-cultures)—with or without SK1-I inhibitor (10 μM)—was added to the wells. The cultures were checked daily via microscopy, and MM viability and cell number were analyzed by luminometry using a Gaussia luciferase assay (New England Biolabs, #E3308), per the manufacturer’s protocol, and a BioTek Synergy H1 automatic plate reader (Winooski, VT, USA).

### 2.9. Lipid Quantification by LC-MS/MS

Lipid extraction was performed using a modified Bligh and Dyer method [44] by suspending cells in chloroform-methanol-2% formic acid with a ratio of 0.5:1:0.4 (v/vo/v) in glass tubes. Internal standards (d7-sphingosine, D7-sphingosine-1-phosphate, and d7-ceramide (N-palmitoyl-D-erythro-sphingosine (d7), all from Avanti Polar Lipids, Birmingham, AL, USA) were added at this stage. The lipid phase was separated by the addition of chloroform and 2% formic acid (1:1, *v*/*v*), which was followed by centrifugation (2500× *g* 15 min). The lower organic phase was saved, and the total phospholipid content was quantified. Sphingolipid analyses were performed via combined liquid chromatography–tandem mass spectrometry (LC-MS/MS), using AB Sciex 6500 QTRAP mass spectrometer with Shimadzu Nexera X2 UHPLC front end (AB Sciex, Framingham, MA, USA) and Ascentis Express RP-Amide column (2.7 μm 2.1 × 50 mm) with gradient elution from methanol:water:formic acid (50:50:0.5, 5 mM ammonium formate) to methanol:chloroform:water:formic acid (90:10:0.5:0.5, 5 mM ammonium formate). Sphingosine-1-phosphate (S1P) was detected in negative ions as *bis*-acetylated derivative as previously described [45]. Ceramides and free sphingoid bases were detected in positive ions. Sphingolipids were normalized by the total lipid phosphorus (Pi) for quantifications in cell extracts and by volume (mL) for quantifications in supernatants.

### 2.10. Real-Time Quantitative RT-PCR (qPCR)

Total mRNA was extracted using RNeasy (QIAGEN, Germantown, MD, USA), per the manufacturer’s protocol, and reverse-transcribed using a High-Capacity cDNA reverse transcription kit (Applied Biosystem, Foster City, CA, USA) on a T100 Thermal Cycler (Bio-Rad Laboratories, Hercules, CA, USA). Quantitative PCR was performed on an CFX96 Real-Time System (Bio-Rad Laboratories, Hercules, CA, USA) using a SsoAdvanced SYBR Green Supermix (Bio-Rad Laboratories, Hercules, CA, USA) and the cDNA equivalent to 40 ng of RNA in a 10 μL reaction, according to the manufacturer’s instructions. The DNA sequences of the human primers used for qPCR are listed in Appendix A. The relative expression was calculated using the comparative 2^−ΔΔCt^ method, with 18S rRNA used as a housekeeping gene.

### 2.11. ChIP-qPCR

ChIP was performed using a modification of the Millipore/Upstate protocol (MCPROTO407). Following the treatments, a total of the 1 × 10^7^ MM.1S or OPM-2 cells in an RPMI-1640 medium were fixed in 1% formaldehyde (Fisher, F79-500) for 10 min at room temperature. Cross-linking was inhibited by the addition of glycine (Fisher, G46-1) to a final concentration 0.125 M. The cell pellets were washed twice with ice-cold PBS and resuspended in an SDS lysis buffer (1% SDS, 10 mM EDTA, 50 mM Tris, pH 8.1), which was supplemented with a protease inhibitor cocktail (Sigma, P8340) and 1 mM of phenylmethylsulfonyl fluoride (Sigma, 93482). The samples were sonicated to generate DNA fragments of 200-bp or 500-bp average lengths (as indicated) on ice using a Fisher Scientific 120 sonic dismembrator (Fisher Scientific, FB-120) as follows: 30 s on and 30 s off at 50% amplitude for 14 cycles on ice and centrifuged at 12,000 rpm for 10 min for 200-bp fragmentation and 30 s on and 30 s off at 50% amplitude for 8 cycles on ice for 500-bp fragmentation. Chromatin was diluted 6-fold in a ChIP dilution buffer (0.01% SDS, 1.1% Triton X-100, 1.2 mM EDTA, 16.7 mM Tris-HCl, pH 8.1, 167 mM NaCl. The total equivalence of the cells were used for each immunoprecipitation. The supernatants were incubated at 4 °C overnight with antibodies to GFI1 (Santa Cruz sc-376949 X) and control IgG (Sigma Aldrich 12-371B). Aliquots for Input chromatin were included with each experiment. The samples were precipitated using 25 μL of Magna ChIP protein A+G magnetic beads for the 200-bp fragmentation and 20 μL for the 500-bp fragmentation (EMD Millipore, 16-663) at 4 °C for 4 h, and subsequently washed once with the following solutions respectively: low-salt buffer (0.1% SDS, 1% Triton X-100, 2 mM EDTA, 20 mM Tris-HCl, pH 8.1, 150 mM NaCl), high-salt buffer (0.1% SDS, 1% Triton X-100, 2 mM EDTA, 20 mM Tris-HCl, pH 8.1, 550 mM NaCl), LiCl wash buffer (0.25 M LiCl, 1% Igepal-CA630, 1% deoxycholic acid, 1 mM EDTA, 10 mM Tris, pH 8.1), and twice with TE buffer (10 mM Tris-HCl, 1 mM EDTA, pH 8.0). The immunocomplexes were then eluted for 4 h at 65 °C with 250 μL of ChIP elution buffer (1% SDS, 0.1 M NaHCO3). To reverse the cross-linking, the eluted samples were treated with 10 μL of 5 M NaCl and incubated at 65 °C for ≥4 h. The DNA was purified using a GeneJET PCR purification kit (Thermo Scientific, K0702). The primer sequences used for ChIP analysis are indicated in Appendix A. qPCR reactions (20 μL) containing 2 Maxima SYBR Green/ROX qPCR Master Mix (Thermo Scientific, K0223), 300 nM of primers, and 4 μL of precipitated DNA for the 200-bp fragmentation and 3 μL for the 500-bp fragmentation were set up in Fast 96-well reaction plates (Applied Biosystems, 4346907). The qPCRs were carried out in a StepOnePlus Applied Biosystems real-time instrument (Thermo Fisher, 4376600). The fold enrichment was calculated based on Ct as 2(ΔCt), where ΔCt = (Ct Input − Ct IP). The enrichment values were adjusted by subtraction of the nonspecific IgG antibody binding and plotted as a relative enrichment. The predicted putative GFI1 binding sites were found using separate searches for GFI1 core AA(T/G)C, GFI1 Jaspar (http://jaspar.genereg.net/about/) accessed on 12 September 2020, and GFI1 Transfac (https://genexplain.com/transfac/) accessed on 22 May 2019.

### 2.12. Statistical Analyses

Statistical analyses were performed using GraphPad Prism 9 software (Irvine, CA, USA). The differences between the groups were compared using a two-tailed unpaired Student’s *t*-test or ANOVA as indicated. A statistically significant difference was set at *p* < 0.05 and the results are expressed as Mean ± SEM. The representative data from at least three biologic replicates are shown.

## 3. Results

### 3.1. Transcriptional Expression Levels of Enzymes That Control S1P Biosynthesis Correlate with GFI1 Levels in MM Cells

To determine the potential of GFI1-dependent modulation of sphingolipid metabolism to promote survival of MM cells, we evaluated the transcriptional expression of genes coding for the main enzymes that control intracellular S1P levels, SphK1 and SGPP1 (see schematic Appendix A), in MM cell lines with altered *GFI1* expression. We found that *SphK1* mRNA levels were downregulated when *GFI1* was knocked-down (KD) by a doxycycline-inducible *shGFI1* construct in both MM.1S ishGFI1 (p53 WT) and OPM-2 ishGFI1 (p53 mut), whereas *SGPP1* mRNA was significantly upregulated when GFI1 levels were decreased (Figure 1A,B left panels).

The opposite was true when these genes were evaluated in the same cell lines that were overexpressing GFI1 (GFI1 o/e) (Figure 1A,B, right panels). This expression pattern of GFI1-dependent elevated *SphK1* mRNA (Appendix A) and decreased *SGPP1* mRNA (Appendix A) was confirmed in other MM cell lines (H929 (p53 WT), KMS-11 (p53 null), and RPMI-8226 (p53 mut)) overexpressing *GFI1* versus their controls. Of note, neither *SphK2* mRNA (Appendix A, left panel) nor *SGPL1* mRNA (Appendix A) levels were altered by *GFI1* overexpression in MM.1S (p53 WT), H929 (p53 WT) or KMS-11 (p53 null) MM cells. We also found that *SphK2* mRNA was not affected by GFI1 overexpression in the JJN3 (p53+/−) and OPM-2 (p53 mut) cell lines (Appendix A, left panel), nor by GFI1 knockdown in the H929 (p53 WT), JJN3 (p53+/−), or OPM-2 (p53 mut) cell lines (Appendix A, right panel). We next evaluated the correlation between endogenous GFI1 levels and SphK1 and SGPP1 in several MM cell lines with differing p53 status. We found that the active form of SphK1 protein (phospho-SphK1) is highly expressed in MM cell lines (Figure 1C left panel) and positively correlates with GFI1 protein levels (r = 0.74) (Figure 1C right panel), while *SGPP1* mRNA displayed a negative correlation (r = −0.961) with *GFI1* mRNA in five different MM cell lines (Figure 1D). Consistent with the mRNA results, SGPP1 immunostaining revealed a lower protein expression in both MM.1S and OPM-2 GFI1 o/e cells when compared with their respective EV controls (Figure 1E). These observations demonstrate a GFI1-dependent dysregulation of the sphingolipid metabolism pathway, which controls S1P levels by activation of S1P synthesis (SphK) and inhibition of S1P degradation (SGPP1) (see schematic in Appendix A).

### 3.2. SphK1 and SGPP1 Expression Divergently Correlate with Increased MM Progression

To confirm our observations in vivo, we analyzed *SphK1* and *SGPP1* levels in MM patient samples and previously published datasets. We analyzed CD138+ cells isolated from normal donors and patients diagnosed with MM (Appendix A) for *SphK1* mRNA and protein levels. We found a trend for increased *SphK1* mRNA levels expressed in MM cells from patients (but not statistically significant) when compared to plasma cells from normal donors (Figure 2A left panel), and P-SphK1 protein levels were highly expressed in CD138+ cells from MM patients (Figure 2A right panel). However, a transcriptome analysis of the GSE6477 dataset did not reveal significant differences in *SphK1* mRNA expression with increased stages of the disease when compared to normal donors’ values (Figure 2B left panel). Also, analysis of the GSE4581 dataset from the Arkansas myeloma database for the association of S1P metabolizing genes and outcome did not show a significant association of higher *SphK1* with shorter survival (Figure 2B right panel).

In contrast, analyzing gene expression arrays from the GSE6477 dataset of different stages of MM, we found progressive and significant repression of *SGPP1* expression compared to normal plasma cells, and showed more profound inhibition of *SGPP1* expression in relapsed MM patients (Figure 2C). Moreover, analysis of the GSE4581 dataset from the Arkansas myeloma database showed that, although there was no difference in survival through the first 29 months, afterwards (particularly from 34 months on), as the patients’ disease progressed, there was a strongly significant association of high *SGPP1* levels with longer survival (*p* = 0.008; HR = 0.1506; n = 121) (Figure 2D). Similar analyses using these data sets for *SGPL1* mRNA did not find any effects on high versus low *SGPL1* levels on overall survival (GSE4581) (D.L.G., UPMC Pittsburgh, PA, USA, personal communication, 2021). These results suggest that changes in sphingolipid metabolism may be clinically significant in MM and support our in vitro findings and overall hypothesis. Together, these changes should increase intracellular S1P, thereby suggesting that changes in sphingolipid metabolism occur during, and may contribute to, disease progression.

### 3.3. Microenvironmental Factors Induce MM Cell Survival through GFI1-Dependent Increase in Intracellular S1P

Multiple factors can increase MM cell survival, growth, and chemoresistance [46]. We recently reported that IL-6, adhesive interactions between MM cells and bone marrow stromal cells (BMSC), and extracellular addition of S1P induce GFI1 expression in MM cells [12]. We also showed that c-Myc levels correlate with GFI1 levels in MM [12]. Therefore, we tested if these factors are also modulating SphK1 and c-Myc levels in MM cells. IL-6 treatment, apart from inducing *GFI1* expression, simultaneously increased *SphK1* and *c-Myc* mRNA levels in H929 (p53 WT) (Figure 3A) and U266 (p53 mut) (Figure 3B) cells, as well as their protein levels in H929 (p53 WT) control (shCtl) cells (Figure 3C) and KMS-11 (p53 null) control (EV) cells (Figure 3D). The P-SphK1 and c-Myc protein levels were lower in IL-6 treated *GFI1* knockdown H929 cells compared to IL-6 treated control cells (Figure 3C). However, the incomplete *GFI1* knockdown in *shGFI1* cells allowed IL-6 induction of GFI1, P-SphK1, and c-Myc (Figure 3C). In KMS-11 (p53 null) cells, overexpression of GFI1 enhanced the increase of P-SphK1 by IL-6 but had little effect on IL-6 induction of c-Myc (Figure 3D).

The adhesive interactions with stromal cells recapitulated these positive effects on *SphK1* and *c-Myc* in H929 (p53 WT) and JJN3 (p53 +/−) cells at the transcriptional levels (Figure 3E), as well as the protein levels in H929 shCtl cells (Figure 3F). As seen with IL-6, the P-SphK1 and c-Myc protein levels were lower in BMSC-treated H929 (p53 WT) *shGFI1* knockdown cells compared to BMSC-treated H929 (p53 WT) control cells (Figure 3F). There is some BMSC induction of GFI1, and hence c-Myc, in the shGFI1 cells compared to the untreated shGFI1 cells, likely due to the incomplete knockdown. Thus, these data strongly suggest that the effects of both of these microenvironmental viability driver signals on the MM cells rely on GFI1 to mediate their effects via S1P-c-Myc increases in MM cells. Exogenous S1P treatment showed a very rapid and dose dependent upregulation of GFI1, P-SphK1, and c-Myc protein levels in both MM.1S (p53 WT) (Figure 3G left panel) and OPM-2 (p53 mut) cells (Figure 3G right panel). Of note, *SphK2* mRNA levels were not changed in MM cells by either IL-6 or co-culture with BMSC (D.N.P., IUPUI Indianapolis, IN, USA, personal communication, 2018). Taken together, these data suggest that pro-survival environmental factors trigger a GFI1-dependent increase of SphK1 and c-Myc levels in MM cells.

### 3.4. GFI1-Dependent Survival of MM Cells Is Mediated by Intracellular S1P Levels

Next, we used several approaches to manipulate intracellular S1P levels in MM cells to investigate if these levels are critical for MM cell viability. First, there was specific pharmacological inhibition of SphK1 with SK1-I, which dose-dependently inhibited the viability of three MM cell lines, regardless of their p53 status (Figure 4A). Moreover, this inhibition was countered by increased GFI1, since overexpression of GFI1 in all cell lines (MM.1S, OPM-2 and KMS-11) provided significant resistance to SK1-I-induced cell death (Figure 4A).

To further confirm the role of intracellular S1P in MM cell survival, we investigated if microenvironmental signals, such as adhesive interactions with stromal cells—shown above to induce GFI1-dependent upregulation of P-SphK1 (Figure 3F)—or osteocytes—previously reported to support MM cell proliferation [47]—can prevent SK1-I induced MM cell death. We used a 3D co-culture model between MM cells (MM.1S) expressing secreted *Guassia* luciferase and the mouse pre-osteoblast stromal cell line MC3T3-E1 MC4 clone or the osteocyte-like cell line MLOA-5. The direct interaction with either pre-osteoblast stromal cells (Figure 4B left panel) or with osteocyte-like cells (Figure 4B right panel) significantly rescued the MM.1S cell viability after SK1-I inhibition for 48 or 72 h. Second, inhibition of SPNS2, a specific S1P transporter that facilitates S1P extracellular export, was used to increase intracellular S1P levels in MM cells. Both pharmacological inhibition by calcitonin (CT in KMS-11 cells) (Figure 4C left panel) or transcriptional inhibition using a stable doxycycline-inducible shSPNS2 MM.1S cell line (Figure 4C right panel) significantly increased the viability of these MM cells. Finally, we increased intracellular S1P by either upregulating its synthesis with a SphK1 pharmacological stimulator (K6PC5) or by reducing intracellular S1P through irreversible degradation with a pharmacological inhibitor of S1P lyase (THI) in MM.1S *GFI1* KD (Figure 4D left panel) and OPM-2 *GFI1* KD (Figure 4D right panel). Both of these treatments significantly improved the viability that was impaired by GFI1 depletion in these cells and had a smaller effect on the GFI1 replete control cells. Taken together, these data support the concept that MM cell viability depends on the maintenance of high intracellular S1P levels, which are positively modulated by GFI1.

### 3.5. Mechanism of Intracellular S1P-Dependent Survival of MM Cells

#### 3.5.1. GFI1 Represses SGPP1 Gene Transcription in MM Cells to Maintain a High Intracellular S1P:Cer Ratio

Our data indicate a new role for GFI1 as a regulator of SL metabolism in MM cells, in part by downregulating the *SGPP1* gene, and is supported by published microarray expression profiling data from murine ES-derived hemogenic endothelial cells that listed *SGPP1* among the 31 genes bound by GFI1 that were upregulated by inhibition of the lysine-specific histone demethylase 1 (LSD1/KDM1A) [48], which is among the repressive chromatin modifiers that GFI1 can recruit [49]. This suggests that modulation of the S1P signaling pathway was significantly associated with genes bound by GFI1, although these were not B cell lineage cells. An ENCODE analysis of DNase hypersensitivity for 95 cell types across the genome and use of an ENCODE algorithm that predicts candidate Cis-Regulatory Elements together revealed multiple putative regulatory regions within the promoter and along ~3 Kb of the 5′ end (exon 1 and a small part of intron 1) of the ~43.6 Kb *SGPP1* gene body [50,51,52,53,54,55,56,57,58,59,60,61] (Appendix A). Thus, to scan for GFI1 occupancy on the *SGPP1* gene in MM cells over a large region (~3 Kb) of the *SGPP1* locus (Figure 5A), we performed an initial broad-resolution GFI1 ChIP-qPCR screen using a 500 bp chromatin fragmentation of fixed chromatin from MM cell lines MM.1S (p53 WT) and OPM-2 (p53 mut) GFI1 o/e and their EV controls (Appendix A).

GFI1 occupancy on the h*ID1* gene at amplicon +275 (+275 nucleotides downstream of the transcription start site (TSS)) was used as a positive control [8] and +66065 nucleotides downstream from the h*RUNX2* TSS was used as a negative (background) control [62]. This initial scan detected GFI1 binding on *SGPP1* in both GFI1 overexpressing MM cell lines using the +661 and +1168 amplicons (midpoints relative to the *SGPP1* TSS), which was significantly higher than the +66,065 h*RUNX2* amplicon and on the same order as the +275 h*ID1* amplicon (Appendix A). We next performed a high-resolution ChIP-qPCR scan with a 200-bp chromatin fragmentation to detect GFI1 occupancy on SGPP1 around the regions of highest GFI1 occupancy in the initial screen (Figure 5A, Appendix A). Under these conditions, significant GFI1 occupancy was detected in both the EV control and GFI1 o/e OPM-2 and MM.1S cells, with enhanced occupancy in cells with GFI1 o/e (Figure 5B). GFI1 occupancy was detected most strongly using the +1108 amplicon (~400-bp into intron 1) for both cell types, with lower, but still significant, GFI1 occupancy detected by the +661 amplicon (near the 3′ end of exon 1). In addition, in the MM.1S cells, but not OPM-2 cells, GFI1 bound fragments were detected by the +964 and the +1225 amplicons. Taking into account that the +1325 amplicon PCR was negative in both cell types and the detection ranges for each amplicon on 200-bp fragments (Figure 5A, Appendix A), a pair of putative GFI1 sites at +1086 and +1109 are within the overlap of the detection ranges for the +964, +1108, and +1225 amplicons (Figure 5A, Appendix A). Three more putative sites (+967, +1133, and +1152) would be detected by two of these three amplicons. While differences exist between the two MM cell types, these data indicate that, regardless of p53 status, GFI1 binds *SGPP1* in two distinct regions in MM cells and this binding is enhanced with increased expression of GFI1.

To validate the functional effect of GFI1-mediated *SGPP1* repression, intracellular levels of different sphingolipid species were measured by mass spectrometry (LC-MS/MS) in MM.1S GFI1 o/e and their respective empty vector (EV) controls. The GFI1 o/e cells had significantly higher intracellular S1P levels when compared to their controls (Figure 5C), while the sphingosine (SPH) levels and total ceramide (Cer) levels were not significantly affected (Figure 5D). Furthermore, the extracellular S1P levels were also elevated in media from GFI1 o/e cells (Appendix A), indicating that the S1P was being exported before it could be either converted back to sphingosine by SGPP1 or degraded by the S1P lyase SGPL1.

#### 3.5.2. Intracellular S1P Levels Modulate c-Myc Protein Levels in a Protein Phosphatase 2A (PP2A)-Dependent Manner

Univariant analysis of the IA15 myeloma dataset from the MMRF database for the association of *GFI1* gene expression and overall survival showed that high *GFI1* was significantly associated with shorter overall survival (Appendix A). Since manipulation of GFI1 levels revealed a correlated modulation of c-Myc (Figure 3C,D,F), and c-Myc is considered a major contributor to the MM malignant phenotype [63,64], we further determined if GFI1-modulated S1P levels control MM cell viability through modulation of c-Myc expression and stability. Treatment of MM cells with the SphK1 inhibitor SK1-I decreased SphK1 active protein levels, as expected, and this was accompanied by dose-dependent diminished protein levels of c-Myc in MM cells (MM.1S, OPM-2 and KMS-11), regardless of their p53 status (Figure 6A). To confirm that SK1-I was acting as expected on S1P levels, we analyzed intracellular S1P and total ceramide levels and found that 10 μM of SK1-I decreased S1P by ~70% and increased total ceramide ~2-fold in MM.1S cells (Appendix A) In contrast, increasing intracellular S1P by specific S1P lyase pharmacological inhibition with THI (Figure 6A schematic) dose-dependently increased c-Myc protein levels in MM.1S and OPM-2 cells (Figure 6B). Together, these data suggest that intracellular S1P levels can regulate c-Myc protein stability. To support these observations, we found a significant inverse correlation between *SGPP1* and *c-Myc* gene expression (Figure 6C) through a gene correlation analysis using the GSE4581 study of newly diagnosed patients from the Arkansas database. Recently, it has been shown in a lung cancer model that ceramide can mediate c-Myc degradation by binding I2PP2A/SET, one of the endogenous PP2A inhibitors [65].

Since PP2A is a known negative regulator of cellular proliferation, we hypothesized that GFI1-dependent modulation of the S1P:Cer ratio towards increased intracellular S1P prevents PP2A activation, and thus stabilizes c-Myc. SK1-I treatment induced a dose-dependent activation of PP2A in both MM.1S and OPM-2 cells as shown by the decreased phosphorylated PP2A levels and was accompanied by a dose-dependent decrease of I2PP2A and P(S62)c-Myc, the stable isoform of c-Myc (Figure 6D). Okadaic acid (OA), a specific inhibitor of PP2A activation, rescued P-PP2A protein levels concomitantly with those of P(S62)c-Myc (Figure 6D).

To assess if modulation of PP2A activity in MM cells is GFI1-dependent, we measured the PP2A activity in OPM-2 ishGFI1 cells. PP2A activity was significantly upregulated in the *GFI1* KD cells (Figure 6E), which had approximately 35% less *GFI1* mRNA levels versus control cells. The specific activation of PP2A was proven with OA pre-treatment, which significantly blunted PP2A activity levels in both control and doxycycline-induced *GFI1* KD OPM-2 cells (Figure 6E). The GFI1-mediated inhibition of PP2A activity was due to increased I2PP2A binding, since the catalytic unit of PP2A (PP2Ac) showed reduced binding with its inhibitor I2PP2A through a co-immunoprecipitation assay in *GFI1* KD OPM-2 cells (Figure 6F left panel). Moreover, co-immunoprecipitation with PP2Ac showed a significant decrease in I2PP2A binding in the MM.1S EV control treated with SK1-I compared to the MM.1S EV control, which was untreated or treated with THI (Figure 6F right panel). Unexpectedly, we were not able to detect an increase in PP2Ac-I2PP2A binding in the MM.1S GFI1 o/e cells (Figure 6F right panel). Consistent with these results, we found that *I2PPA/SET* mRNA levels are significantly increased with MM disease progression (Appendix A). In support of the capacity of OA pre-treatment to prevent SK1-I-induced loss of c-Myc stability, OA pre-treatment also significantly rescued both MM.1S and OPM-2 cell viability from the effects of 48 h SK1-I treatment (Figure 6G), as well as RPMI-8266 (p53 mut) cell viability (Appendix A left panel). These findings were also validated by modeling paracrine increases (outside-in) of palmitoyl ceramide (Cer16:0) on MM cells, as a surrogate for intracellular Cer accumulation [66]. RPMI-8226 (p53 mut) and MM.1S (p53 WT) cells both exhibited marked decreases in viability (Appendix A) when challenged with ceramide Cer16:0 and compared with vehicle control. Although the p53 mutant cell line RPMI-8266, as well as MM.1S GFI1 o/e cells, were more resistant to both Cer and SK1-I treatments, in all cases the viability was significantly restored by OA pre-treatment (Figure 6G and Appendix A). Cells exposed to exogenous ceramide trigger further endogenous ceramide production via the de novo synthesis pathway [66]. Therefore, similar to the effects induced by increased intracellular Cer levels resulting from SK1-I treatment, the exogenous Cer16:0 treatment decreased c-Myc and P-PP2A protein levels, which were restored by OA pre-treatment in MM.1S (Appendix A left panel) and OPM-2 (Appendix A right panel). To confirm if c-Myc protein modulation by GFI1 depends on intracellular S1P levels, we increased intracellular S1P levels in dox-induced *GFI1* knockdown cells (MM.1S and OPM-2 ishGFI1) and their untreated controls by treatment with either K6PC5 or THI. THI, but not K6PC5, treatment partially restored c-Myc protein levels after only 4 h in both cell lines, as did inhibition of PP2A with OA (Figure 6H). Since K6PC5 rescued the viability of these cells at 48 h better than THI (Figure 4D), the inability to detect c-Myc restoration at 4 h with K6PC5 may be a function of the time point we selected. However, the ability of OA treatment to restore c-Myc stability after 4 h is reflected in the finding that OA treatment for 48 h dose-dependently and significantly rescued viability of both the MM.1S (Figure 6I upper panel) and OPM-2 dox-induced *GFI1* knockdown cells (Figure 6I lower panel). Interestingly, restoration of c-Myc induced by the forced increase in intracellular S1P was accompanied by partial recovery of P-SphK1 and GFI1 protein levels (Figure 6B,H), indicating the activation of a positive feedback loop. Pitson et al. reported that the rapid and transient increase in SphK1 activity is a result of the activating phosphorylation at Ser225 in human SphK1 by phospho-extracellular signal-regulated kinases 1 and 2 (P-ERK1/2) [67], which are crucial for subsequent oncogenic signaling by this enzyme. Importantly, PP2A has been reported to be the phosphatase that dephosphorylates phospho-Ser225-SphK1 to deactivate SphK1 [68]. The induced intracellular S1P increase by THI significantly activated ERK1/2 in the presence, and more so in the absence, of GFI1, which might be responsible for the SphK1 reactivation, and this could contribute to the observed c-Myc recovery in both *GFI1* knockdown MM.1S and OPM-2 cell lines (Appendix A). Notably, K6PC5, which was not able to rescue c-Myc at 4 h, was also unable to boost ERK1/2 activation in response to decreased GFI1 in *GFI1* knockdown MM.1S, although there was some increase in *GFI1* knockdown OPM-2 cells. Exogenous S1P induced a very rapid dose-dependent activation of ERK1/2 and SphK1 in both MM.1S and OPM-2 doxycycline-treated ishGFI1 cells and their controls (Appendix A). As a result, PP2A was inactivated in the *GFI1* knockdown cells and c-Myc levels were restored. To confirm GFI1-dependent inactivation of PP2A, OA treatment of *GFI1* knockdown cells restored P-ERK1/2 (Appendix A) and P-SphK1 (Figure 6H) levels in both MM.1S and OPM-2 cell lines. Taken together, these data suggest that GFI1 represses the expression of SGPP1, thus maintaining high intracellular S1P levels that keep PP2A inactive, thereby leading to high c-Myc protein levels.

### 3.6. Decreased Intracellular S1P Levels Induce Cell Death via Autophagy and Reveals That GFI1-Protection of MM Cells Viability via S1P and p53WT Are Independent Pathways

Since SK1-I treatment of MM cells had a profound and dose-dependent inhibitory effect on their viability (Figure 4A), we investigated what type of cell death is triggered by the deregulation of the S1P:Cer ratio. SK1-I treatment failed to induce caspase 3 activation in either MM.1S (p53 WT) (Figure 7A left panel) or OPM-2 (p53 mut) cells (Figure 7A right panel), regardless of their GFI1 expression levels. The lack of caspase 3 activation by SK1-I was confirmed by the inability of the caspase 3-specific inhibitor (Z-DEV-D-FMK) to rescue viability after SK1-I treatment in both these cell lines (D.N.P., IUPUI Indianapolis, IN, USA, personal communication 2020). Similar with Z-DEV-D-FMK, a necroptosis blockade by Necrostatin 1 (a RIP1K specific inhibitor) had no effect on the SK1-I ability to decrease MM cell viability (D.N.P., IUPUI Indianapolis, IN, USA, personal communication 2020). In contrast, SK1-I treatment induced a dose-dependent activation of autophagy in both MM cell types, as shown by the conversion, via lipidation, of microtubule-associated protein 1 light-chain 3 (LC3) from LC3-I (free form) to LC3-II (membrane-bound form) (Figure 7A both panels). Therefore, we tested if the upregulated autophagic flux we detected is responsible for the observed decreased viability using pre-treatment with Bafilomycin (Baf), a known inhibitor of autophago-lysosome formation as proven by the blocked autophagic flux observed in Figure 7B (increased LC3II/I ratio coupled with increased p62 levels). By itself, Bafilomycin induced a loss of viability in both MM.1S (Figure 7C left panel) and OPM-2 cells (Figure 7C right panel). However, Bafilomycin prevented SK1-I from inducing a further decrease in MM viability. Similarly, by itself, Bafilomycin decreased the level of P(62)-c-Myc, but then protected the cells from SK1-I inducing further loss of P(62)-c-Myc (Figure 7B). Interestingly, the viability boost by GFI1 overexpression (+/− SK1-I) was lost in Bafilomycin-treated p53 mut OPM-2 cells, but not in Bafilomycin-treated p53 WT MM.1S cells. Together, these results suggest that induction of cell death by low intracellular S1P requires autophagy. Further, GFI1 enhancement of viability in p53 WT MM.1S cells through inhibition of p53 function [12] still occurs even when low S1P has no effect on viability. In contrast, p53mut OPM-2 cells GFI1 enhancement of viability is lost when low intracellular S1P has no effect on viability due to the presence of Bafilomycin. In these cells, p53 is not functional and therefore can’t be inhibited by GFI1. This indicates that GFI1 protection of MM cell viability via the regulation of p53 function and S1P levels are independent mechanisms.

## 4. Discussion

Our recent publication demonstrated an important role for the transcriptional repressor GFI1 in promoting MM cell survival and growth as well as increasing bone destruction and contributing to bortezomib resistance [12]. In addition to describing a mechanism in p53-WT MM cells, in this publication we noted that GFI1 also may play a role in the survival and growth of p53-mutant or -null MM cells as indicated by *GFI1* knockdown studies. Although p53 mutations are rare in MM (only 10% of the diagnosed patients) [69], patients harboring the 17p13 deletion [del(17p)] are considered “high risk” and have poorer outcomes and shorter survival times compared to standard-risk patients [70,71,72]. Here we report a novel and previously unexplored mechanistic role for GFI1 as a modulator of sphingolipid metabolism, by which it regulates survival of MM cells regardless of their p53 status (see schematic in Appendix A), and acts in addition to GFI1 inhibition of p53 WT in p53 replete cells.

The link between sphingolipids and myeloma is well recognized in Gaucher disease, an inborn disorder characterized by a deficiency in lysosomal glucocerebrosidase [73]. Gaucher patients have increased risk of MM possibly from: (1) chronic elevation of inflammatory cytokines in the tumor niche; (2) inappropriate balance of sphingolipids, with high S1P and low Cer in the malignant cells [74,75]; and (3) chronic osteoclast activation, which releases growth factors and cytokines from the bone matrix [76] and are also mechanisms relevant in MM bone disease.

Using MM cell lines, we found that the GFI1 expression inversely correlates with *SGPP1* mRNA, as well as protein levels, while it positively correlates with activated SphK1 protein levels, leading to elevated S1P levels. GFI1 overexpression blunted *SGPP1* expression in both p53 WT and p53 mut MM cell lines as compared to controls. Together, these observations reveal a new role for GFI1 as a positive modulator of intracellular S1P levels. Numerous studies have shown that *SphK1* mRNA and protein levels are often upregulated in cancerous tissues, and elevated production of S1P is correlated with chemo- and radio-resistance and poor prognosis [19,77]. However, in the MM datasets that we examined, we did not detect increased *SphK1* mRNA expression with disease progression, nor did we find an association of higher expression with shorter survival in a set of newly diagnosed MM patients. In our own studies, CD138+ cells isolated from a small number of patients showed only a trend towards elevated *SphK1* mRNA levels compared to cells from normal donors. One possibility is that in MM cells there is more activated SphK1 protein (which we observed in 2/3 patient samples) that is not a function of *SphK1* mRNA changes. We previously reported that increased GFI1 expression was associated with disease progression [12] and showed here that high *GFI1* was also significantly associated with shorter overall survival. Since GFI1 is known to be a transcriptional repressor, it’s unlikely to directly promote increased *SphK1* mRNA expression, but may promote increased activation of SphK1 protein function. Importantly, and consistent with elevated GFI1 expression during MM progression, *SGPP1* expression significantly decreased with MM progression, which supports a key role for higher GFI1-dependent intracellular S1P levels in MM cell survival. Analysis of *SGPP1* expression of the same newly diagnosed MM patients showed a significant association of higher levels of *SGPP1* expression with longer patient survival after the first 30 months of the study length, which suggests that in patients, lower S1P levels due to higher SGPP1 activity decreases MM cell viability. Moreover, the SGPP1 enzyme has been recently described as a regulator of sphingolipid metabolism and apoptosis in mammalian systems [78]. All together these data strongly indicate the association of the disease with a high production of intracellular S1P.

S1P secreted by the tumor cells signals through interaction with their transmembrane S1P receptors (S1PRs) in an autocrine manner to promote growth, survival, motility, and metastasis [79,80], or in a paracrine manner to induce angiogenesis and regulate tumor–stromal interactions as well as immune cells [81]. However, intracellular S1P may also promote cancer progression via intracellular targets, such as HDAC1/2 [82] and NF-κB [83], and can promote cancer cell resistance to therapy by counteracting the pro-apoptotic effects of ceramide [84].

We found that microenvironmental factors that enhance MM survival, such as IL-6, S1P, and adhesive interactions with BMSC, and that these also induce GFI1 expression [12] and modulate SphK1 and c-Myc levels in MM cells. These results support the role of GFI1-mediated elevated intracellular S1P in MM cell survival. These observations are consistent with previous reports that found SphK1 played a key role in IL-6 induced myeloma cell proliferation and survival [85,86,87,88]. SphK2 has also been reported to play a supportive role in MM cell survival since SphK2-specific inhibition induced apoptosis in MM cells in vitro and suppressed myeloma tumor growth in vivo in mouse myeloma xenograft models [89]. Further, we demonstrated the role of intracellular S1P in MM survival by manipulating the S1P level in several ways: specific SphK1 inhibition, S1P extracellular transport blockade, SphK1 activity stimulation, and inhibition of intracellular irreversible S1P degradation.

We previously reported that increased levels of GFI1 increased MM cell viability [12], and here we showed that upregulation of GFI1 increased the pro-survival sphingolipid S1P. Since GFI1 often acts as a negative transcription factor that binds genes directly and then recruits chromatin modifiers to alter the chromatin architecture towards repression, we hypothesized that GFI1 upregulates S1P levels by repressing transcription of the gene encoding the phosphatase SGPP1, which de-phosphorylates S1P to form SPH. We have shown that GFI1 overexpression in MM cells enhanced cell growth and reduced levels of *SGPP1* mRNA. GFI1 has a C-terminal DNA binding domain containing six C2H2 Zinc-fingers that bind Gfi1 DNA recognition sites. GFI1 serves as a recruitment nucleator of complexes such as the LSD1 (KDM1A)-CoREST Repressive Complex, Histone Deacetylases (HDACs), and Polycomb Repressive Complex 2 (PRC2), in part, via the GFI1 N-terminal SNAG domain [6]. The recruited chromatin-modifying enzymes alter the histones, inducing a repressed chromatin architecture that results in reduced transcription [10,62,90]. In the current study, we showed that GFI1 directly bound to two regions of the human *SGPP1* gene (a 3′ region of exon 1 and a 5′ region of intron 1) (Figure 5A). Marneth et al. [91] reported a GFI1 ChIP-seq using human Kasumi-1 AML cells. Their data in CistromeDB loaded into a UCSC Genome browser track revealed broad binding peaks over the *SGPP1* promoter and the 5′ half of exon 1 (Appendix A). Interestingly, the two GFI1 binding regions that we identified in the two MM cell lines using ChIP-qPCR are at the trailing edge of the GFI1 ChIP-seq peak in human Kasumi-1 AML cells [91]. This suggests that there may be cell-type specific regulation of the *SGPP1* gene by GFI1. Cell-type specific (and perhaps species-specific) binding by GFI1 is also supported by the location of the reported Gfi1 occupancy region 14 Kb upstream of the *Sgpp1* gene TSS in murine ES-derived hemogenic endothelial cells [48]. The ENCODE Registry of candidate cis-Regulatory Elements (cCREs) predicts that both of the regions we found to bind GFI1 in MM cells are within a set of proximal and distal enhancer-like signatures. This is supported by the presence of DNase hypersensitive sites throughout this region in a B lymphocytic cell line, as well as in the 95 cell lines in the ENCODE project (Appendix A). Published ChIP-seq data for known enhancer regulatory histone marks in MM1S cells obtained from CistromeDB and visualized in the UCSC Genome Browser (Appendix A) reveal histone modifications consistent with the presence of an enhancer region at the *SGPP1* exon 1/intron1 GFI1 binding regions, but not at the promoter. Our identified GFI1 binding sites correlate directly with peaks for H3K1me1 histone modifications in MM.1S, which typically are found in enhancer regions. Additionally, histone modifications (H3K4me3, H3K27ac, H3K27me3, and H3K9ac) typical of either distal- or proximal-enhancer regions or promoter-proximal regions are found throughout exon 1 and at the 5′ end of intron 1 in MM.1S cells. Together, the DNase1 hypersensitivity and predicted ENCODE cCRE’s, along with the distribution of histone modifications found in MM.1S cells, suggests a regulatory region throughout exon 1 and into intron 1, and in particular where GFI1 binds the *SGPP1* gene in MM cells. These data support our findings that GFI1 binds to downstream *SGPP1* enhancer regions, resulting in decreased gene expression—possibly by recruiting histone modifiers and/or impeding the progress of RNA Polymerase II processivity. This is consistent with the increased occupancy of ectopically-expressed GFI1 on the *SGPP1* gene, resulting in reduced SGPP1 mRNA and increased S1P levels. Although we found strong evidence that GFI1 is repressing *SGPP1*, we cannot completely rule out its involvement in also repressing *SGPL1*. Our data show that intracellular (Figure 5C) and extracellular S1P levels (Appendix A) are significantly elevated when compared with their control counterparts, which indicate that S1P lyase might be inhibited. These results might argue the involvement of GFI1 as a repressor of *SGPL1*. However, it is unlikely that this is true since inhibiting S1P lyase with THI clearly improved c-Myc levels and MM cells viability, indicating S1P lyase is an active enzyme in our system, and modifying GFI1 levels didn’t change SGPL1 transcripts in any tested cell line with altered GFI1 expression (Appendix A). Further investigations are needed to clarify this issue.

Both GFI1 [12] and c-Myc [92,93] are increased with disease progression and GFI1 levels correlate with c-Myc levels in MM cells [12]. This, along with the accepted concept of c-Myc addiction in MM [29] and the data from patient arrays showing *SGPP1* gene expression decreasing with the disease progression and inversely correlating with *GFI1* and *c-Myc* gene expression, prompted us to hypothesize that GFI1 transcriptional repression of *SGPP1* contributes to elevated c-Myc in MM cells. The regulation of c-Myc biology is fundamental to maintain a correct balance between normal and aberrant cellular phenotypes. PP2A is a key regulator of many oncoprotein signaling pathways, including c-Myc and its activation—which leads to c-Myc degradation [35]. Proper regulation of c-Myc expression levels during stimulation of cell proliferation is controlled by c-Myc protein stability, which is maintained by phosphorylation at Serine 62 [94]. Recently, it has been shown in a lung cancer model that ceramide can mediate c-Myc degradation through a PP2A (protein phosphatase 2A)-dependent mechanism. The endogenous PP2A inhibitor, I2PP2A (SET), was identified as one of the major ceramide binding proteins and the Cer-I2PP2A interaction competes for the availability of I2PP2A to inhibit PP2A activity, thereby releasing active PP2A to dephosphorylate and destabilize c-Myc [65]. We demonstrated that the survival status of MM cells is controlled by GFI1-dependent dysregulation of the S1P:Cer ratio such that it shifts the ratio towards upregulation of c-Myc stability through increased availability of I2PP2A, leading to inhibition of PP2A.

Our results also showed no caspase 3 activation in MM cells with SK1-I treatment that specifically inhibits SphK1 regardless of the MM cell p53 status. This differs from Venkata et al. who showed that SphK2-specfic inhibition induced caspase 3-dependent cell death in MM cells [89]. Possibly, this may reflect the fact that although both SphK isoforms modulate S1P production, several studies have suggested that their biological roles and their localization are different [17,50,51,95,96].

Finally, we found that SK1-I treatment caused the S1P:Cer ratio to shift towards Cer, which augmented autophagy in MM cells, regardless of their p53 status. Further, the SK1-I induced loss of viability could be blocked by inhibition of autophagy with BAF, indicating that the SK1-I induced autophagy leads to decreased viability. However, the capacity of GFI1 overexpression to increase MM viability was lost in BAF-treated p53 mutant cells, but not in Bafilomycin-treated p53 WT cells. Based on our previous report that GFI1-p53 interaction enhanced MM cell viability by blocking p53-induced apoptosis [12], this result suggests that in p53 replete MM cells, GFI1 is protecting the cells by both inhibiting p53 and increasing S1P. In contrast, in p53 mut cells, GFI1 is only protecting the cells via S1P modulation. Taken together, our data are in agreement with previous work that suggested sphingolipid modulation is a very plausible therapeutic strategy for MM [52].

## 5. Conclusions

In summary, our results support GFI1 as a key contributor to the growth and survival of MM cells through its regulation of sphingolipid metabolism as well as inhibition of p53 activity [12]. GFI1 represses *SGPP1* transcription in MM cells regardless of their p53 status, thus contributing to c-Myc stabilization. This feature supports GFI1 as a very attractive therapeutic target for all types of MM, including the “high risk” patient population with non-functioning p53 that have early relapse and become highly resistant to currently available therapy. Moreover, because GFI1 can modulate the levels of specific bioactive lipid components that can modify cancer cell fate, this suggests that targeting GFI1 may also provide a new therapeutic approach for other types of aggressive cancers expressing high levels of c-Myc.

## Figures and Tables

**Figure 1 cancers-14-00772-f001:**
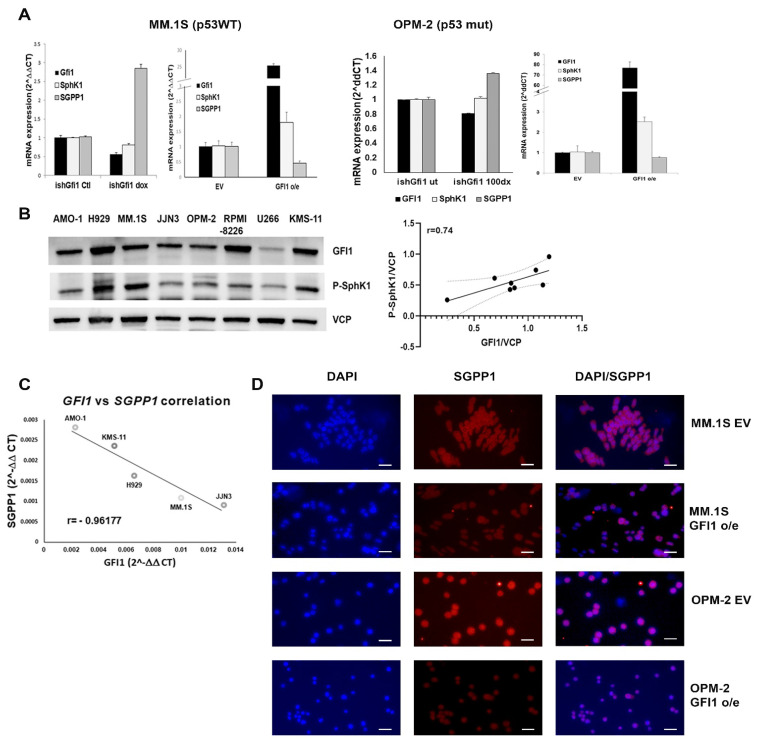
GFI1 modulates the expression of enzymes that control S1P production in MM cells. The mRNA levels of *GFI1*, *SphK1,* and *SGPP1* were measured by qPCR in transduced MM.1S (p53 WT) (**A**) and OPM-2 (p53 mut) (**B**) cells as follows: for GFI1 knockdown (KD), stable ishGFI1 expressing cells were treated for 3 days with 1 μg/mL dox and compared with untreated Controls; for GFI1 overexpression (o/e), controls were empty vector (EV) expressing cells (**A**,**B**); Western blots (WB) showing GFI1 and phosphor-SphK1 (P-SphK1) protein levels in the denoted MM cell lines. Valosin containing protein (VCP) levels were used as loading controls (left panel) and the linear regression between these protein expressions was graphed and the Pearson correlation calculated (r = 0.74) (right panel) (Original data in Appendix A) (**C**); *GFI1* and *SGPP1* mRNA expression were measured by qPCR in five different MM cell lines and the linear regression was graphed and Pearson correlation calculated (r = −0.96177) (**D**); Representative images of paraformaldehyde fixed cytospins of MM.1S and OPM-2 cells (EV and GFI1 o/e) stained for SGPP1 (red-Alexa Fluor^®^ 568) and DAPI for nuclei (Magnification ×20, bar 20 μm) (**E**).

**Figure 2 cancers-14-00772-f002:**
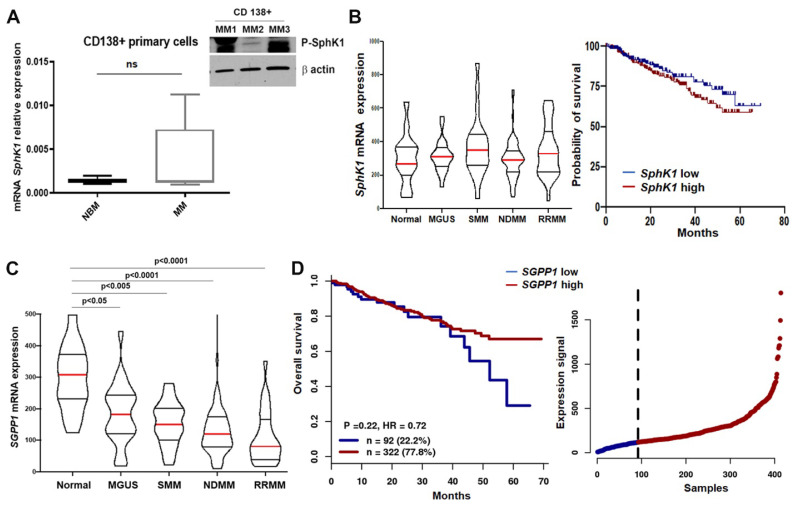
*SphK1* and *SGPP1* expression divergently correlate with MM patients’ progression. The mRNA expression levels of *SphK1* in CD138+ primary cells of MM patients (n = 7; see MM 4-10 in patient/normal donors characteristics Appendix A) compared with normal donors (n = 3; see ND 1-3 in patient/normal donors characteristics Appendix A) (left panel); unpaired *t*-test with Welch’s correction showed differences between NBM vs. MM were not significant (ns; *p* = 0.187); P-SphK1 protein levels as measured by WB (Original data in Appendix A) in (left panel) in three CD138+ primary cells of MM patients (see MM 1-3 in patient/normal donors characteristics Appendix A) (right panel) (**A**); Transcriptome analysis of the GSE6477 dataset showing *SphK1* mRNA expression in different stages of the disease: Monoclonal gammopathy of undetermined significance (MGUS; n = 22), smoldering myeloma (SMM; n = 24), newly diagnosed MM (NDMM; n = 73), and relapsed refractory MM (RRMM; n = 28) were compared to normal (N) (n = 15), (Tukey’s multiple comparison test shows no significance among the 6 groups) (left panel) and *SphK1* mRNA expression analysis of newly diagnosed MM patients (before treatment) (n = 414) using GSE4581 dataset showing probability of survival for low expression (blue, n = 207) and high expression (red, n = 207) (Log-rank test showed no significant difference between groups *p* = 0.3086) (right panel) (**B**); Analysis of mRNA *SGPP1* expression in MM patients using GSE6477 dataset in different stages of plasma cell neoplasm (same groups as in B left panel) using Tukey’s multiple comparison test shows significance between normal and MGUS (*p* = 0.03), normal and SMM (*p* = 0.0015), normal and NDMM (*p* < 0.0001), and normal and RRMM (*p* < 0.0001) (**C**); Overall survival based on *SGPP1* mRNA expression (low in blue, n = 92 and high in red, n = 322) (*p* = 0.22) in newly diagnosed MM patients using Arkansas database and GSE4581 dataset (left panel) and expression signal of *SGPP1* in the patient cohort (right panel) (**D**).

**Figure 3 cancers-14-00772-f003:**
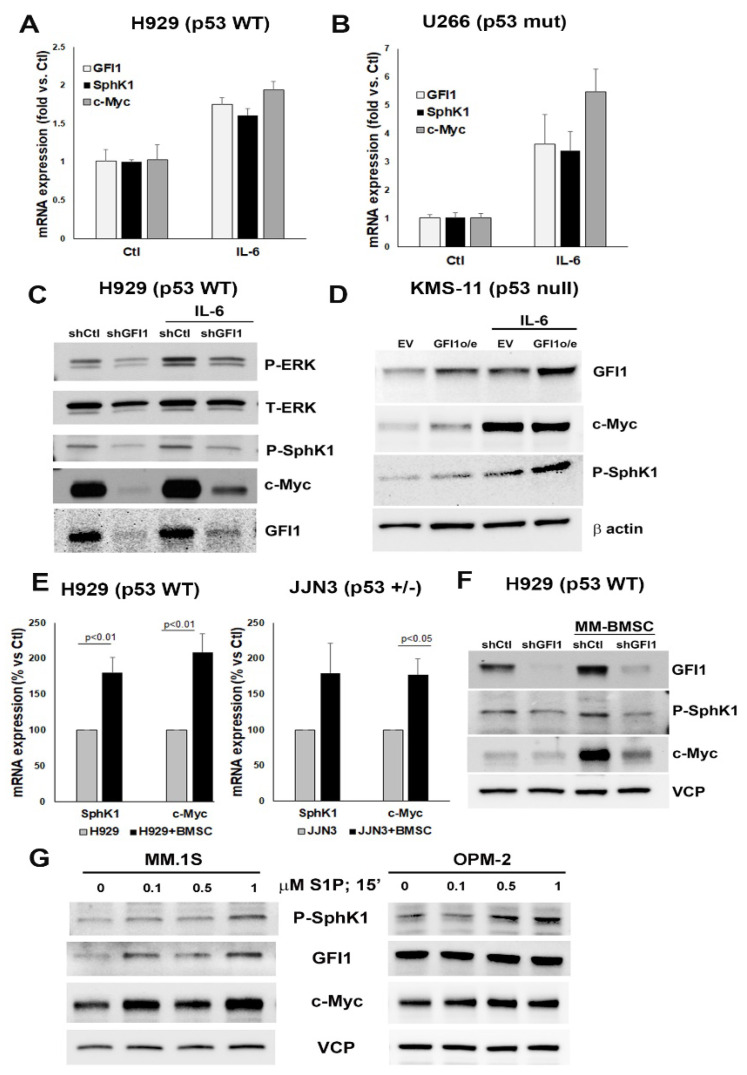
Microenvironmental factors activate the GFI1-SphK1-c-Myc axis for survival. *GFI1*, *SphK1*, and *c-Myc* mRNA expression detected by qPCR and expressed as fold change versus untreated control in MM cell lines H929 (p53 WT) (**A**) and U266 (p53 mut) (**B**) treated or not with IL-6 (5 ng/mL; 4 h), indicated proteins expression levels by WB (using total ERK1/2 (T-ERK1/2) and β actin [see Appendix A] as loading control) in H929 cells transiently knocked down using sh*GFI1* and Sh-vector control (**C**), and in KMS-11 (p53 null) cells stably expressing GFI1 o/e and their EV controls (using β actin as loading control) (**D**) treated or not with IL-6 (5 ng/mL; 4 h); MM cells H929 (p53 WT) (left panel) and JJN3 (p53+/−) (right panel) were directly co-cultured for 24 h with bone marrow stromal cells (BMSC) obtained from MM donors (MM 3, 6, and 9 in Appendix A). MM cells were harvested separately and *SphK1* and *c-Myc* mRNA levels were measured by qPCR and compared with levels in MM cells alone (n = 3) (**E**); GFI1, P-SphK1, and c-Myc protein expression by WB using VCP as loading control in H929 cells transiently knocked down by sh*GFI1* and sh-vector control and co-cultured or not with MM patient derived BMSC (MM3 in Appendix A) (**F**); GFI1, P-SphK1, and c-Myc protein expression by WB in MM.1S (p53 WT) (left panel) and OPM-2 (p53 mut) (right panel) treated for 15 min with the specified concentrations of S1P or vehicle control (ethanol) and normalized by VCP (**G**). Original WB data in Appendix A (**C**,**D**,**F**,**G**).

**Figure 4 cancers-14-00772-f004:**
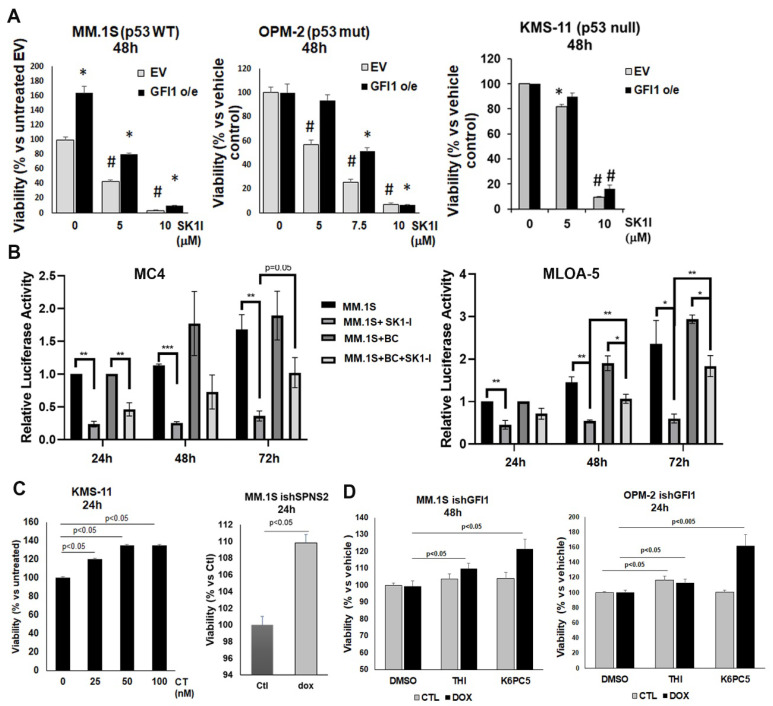
GFI1-dependent survival of MM cells is mediated by intracellular S1P levels. MM.1S, OPM-2, and KMS-11 GFI1 o/e and empty vector (EV) control cells were treated with different doses of SK1-I (5, 7.5 and 10 μM) or vehicle control (H_2_O) (0 μM) for 48 h. Viability was measured by AlamarBlue assay and reported as percent vs. untreated EV control (MM.1S left panel) or percent vs. vehicle control for each cell type to better appreciate the difference between EV and GFI1 o/e, (OPM-2 middle panel and KMS-11 right panel) (* *p* < 0.05 vs. untreated control and # *p* < 0.005 vs. untreated control) (**A**); Mouse pre-osteoblastic MC4 cell line (left panel) and mouse pre-osteocytes MLO-A5 cell line were co-cultured in a 3D-model with MM.1S-gLuc cells in a 1:5 ratio and treated or not with SK1-I (10 μM). Relative secreted *Guassia* luciferase activity in the supernatant was measured after 24, 48, and 72 h by a luminometer (* *p* < 0.05, ** *p* < 0.01 and *** *p* < 0.0001) (**B**); SPNS2 specific S1P transporter was inhibited by either pharmacological treatment with Calcitonin (CT) (0, 25, 50, and 100 nM) of KMS-11 cells (left panel) or transcriptional repression (transduction with doxycycline-inducible shSPNS2 plasmid, 30% inhibition of SPNS2 gene expression in dox-treated vs. untreated controls after three days) in MM.1S cells. Viability was measured by AlamarBlue assay and reported as percent vs. untreated control (**C**); GFI1 knockdown was induced in MM.1S ishGFI1 and OPM-2 ishGFI1 cells by dox treatment (1 μg/mL) for three days. MM.1S ishGFI1 dox treated (40% GFI1 KD) and OPM-2 ishGFI1 dox treated (30% GFI1 KD) were then treated with THI (10 μM), K6PC5 (5 μM), or vehicle (DMSO) for an additional 48 h (MM.1S) or 24 h (OPM-2) and viability was measured by AlamarBlue assay and reported as percent vs. each cell type vehicle control (**D**).

**Figure 5 cancers-14-00772-f005:**
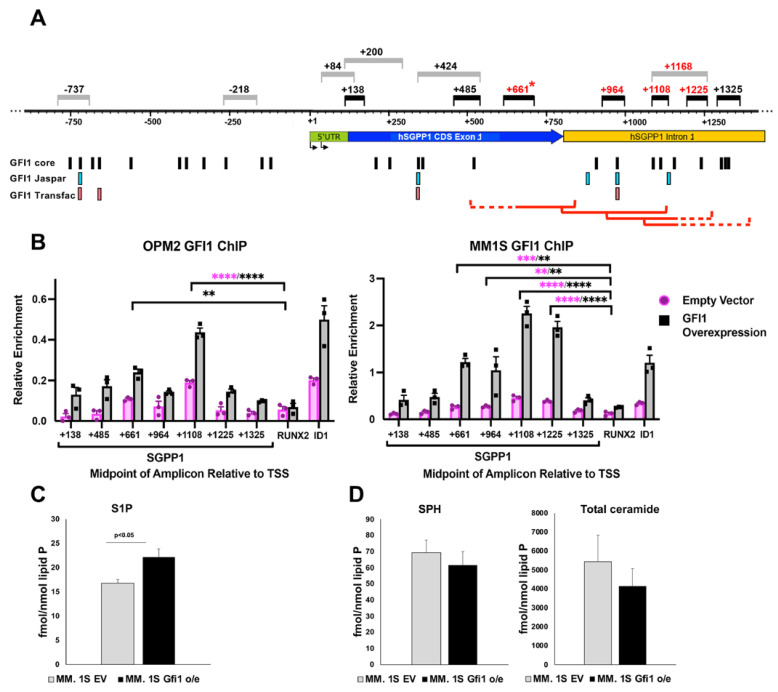
GFI1 ChIP-qPCR studies reveal GFI1 binding on *SGPP1* in MM cells. A schematic of the promoter, exon 1, and part of intron 1 of human *SGPP1* NM_030791.4 (GRCh38.p12; GCF_000001405.27) is presented below a nucleotide scale labeled with respect to the *SGPP1* TSS (+1) at nucleotide 63,728,065. PCR amplicons (midpoints relative to *SGPP1* TSS) used to analyze GFI1 ChIP-qPCR using 500-bp (Gray bars) or 200-bp (Black bars) chromatin fragmentation; red amplicon # has significant results in this study. * +661 amplicon was used in both screens. The predicted putative GFI1 binding sites found using searches for the GFI1 core AA(T/G)C; or the Jaspar; and Transfac algorithms are shown, along with red brackets that indicate the detection ranges of the positive amplicons (dashes; overlaps with the detection range of a negative amplicon). (**A**); Chromatin of OPM-2 (left panel) and MM.1S (right panel) cell lines, GFI1 o/e and EV control, which were fragmented to an average size of 200-bp. Sites on *SGPP1* were compared to a background control amplicon on the *RUNX2* gene (+66,065) and a positive control amplicon on the *ID1* gene (+275). The primer sequences are indicated in Appendix A along with their detection ranges. Data show the mean of three biological replicates with the standard error. Statistical analyses used one-way ANOVA with Dunnett’s multiple comparisons correction comparing to “RUNX2” in the same cells: * *p* < 0.05; ** *p* < 0.01; *** *p* < 0.001; **** *p* < 0.0001. (**B**); Abundance of sphingolipid species: S1P (**C**); sphingosine (SPH) (**D**-left panel) and total ceramide (**D**—right panel) as measured by LC-MS/MS in MM.1S cells GFI1 o/e compared to their EV controls. Data are the average of three biological replicates (Mean +/− SEM).

**Figure 6 cancers-14-00772-f006:**
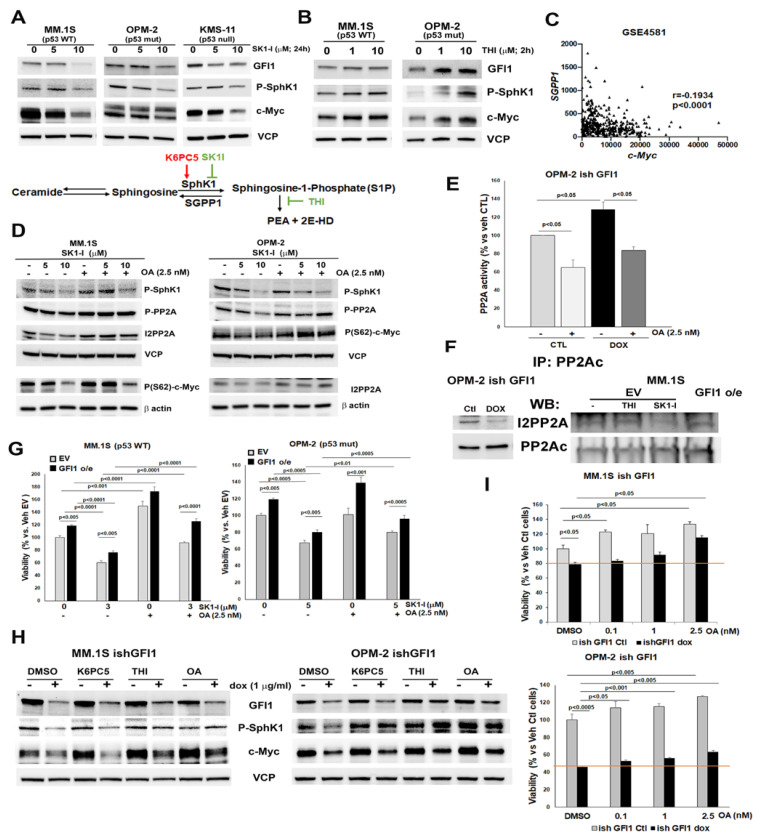
Intracellular S1P levels modulate c-Myc protein stability in a protein phosphatase 2A (PP2A)-dependent manner. Cell lysates were analyzed by WB (original data in Appendix A) using specific antibodies as indicated and representative immunoblots are shown (**A**,**B**,**D**,**F**,**H**). Immunoblots are arranged with the internal control used for the same samples and, therefore, are not in the same order for the different cell lines in (**D**). DMSO is the vehicle control (also shown as 0) for all treatments. MM.1S ishGFI1 and OPM-2 ishGFI1 were treated or not with dox (1 μg/mL, 3 days) to induce GFI1 KD (E, F, H-I). MM.1S, OPM-2 and KMS-11 cells were treated with SK1-I for 24 h. A schematic of S1P synthesis and degradation (main enzymes and their specific inhibitors) is shown below (**A**); MM.1S and OPM-2 cells were treated with THI (0, 1, 10 μM, 24 h) (**B**); Spearman correlations (r= −0.1934) between *c-Myc* and *SGPP1* mRNAs (*p* < 0.0001 by two-tailed test) in MM patient samples (n = 414) of GSE4581 database (**C**); MM.1S and OPM-2 cells pre-exposed to OA (2.5 nM, 2 h) were treated with SK1-I for 24 h (**D**); OPM-2 GFI1 KD and control cells were exposed to OA for 2 h. PP2A activity was measured and results (n = 3) were graphed as percentage activity vs. vehicle control (**E**); MM.1S GFI1 o/e cells and EV treated with either THI (10 μM, 2 h) or SK1-I (5 μM, 2 h) and as well as OPM-2 GFI1 KD and Control cells were IP with anti-PP2Ac antibody and probed for I2PP2A and PP2Ac by WB (**F**); MM.1S and OPM-2 GFI1 o/e and EV control cells pre-exposed to OA (2.5 nM, 2 h) were then treated with SK1-I (3 μM and 5 μM respectively). Viability after 48 h was reported as percent vs. vehicle treated control for each cell type (**G**); MM.1S and OPM-2 GFI1 KD and Control (**H**,**I**) were treated with either K6PC5 (5 μM), THI (10 μM), OA (2.5 nM) or DMSO for 4 h for WB (**H**), or were exposed to OA (0, 0.1, 1, 2.5 nM) and viability after 48 h was reported as percent vs. vehicle treated control for each cell type (**I**).

**Figure 7 cancers-14-00772-f007:**
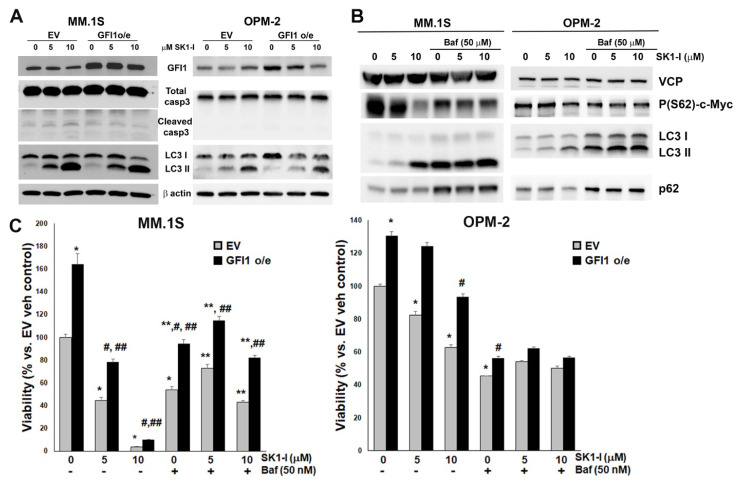
Insufficient intracellular S1P in MM cells induces cell death via autophagy. MM.1S and OPM-2 cells were treated with SK1-I (5 and 10 μM) or vehicle control (DMSO shown as 0) for 24 h (**A**,**B**) or 48 h (**C**) and pre-treated or not with Baf (50 nM, 2 h) (**B**,**C**). Whole cell lysates were probed by WB with indicated specific antibodies and presented by representative immunoblots (**A**,**B**) (original data in Appendix A). Viability was measured by AlamarBlue assay and reported as percent vs. EV vehicle treated control (*p* < 0.0005 in both MM.1S and OPM-2 panels: * vs. vehicle control; ** vs. respective SK1-I treatments alone; # vs. GFI1 o/e vehicle control and ## vs. EV respective treatments) (**C**).

## Data Availability

The anonymized, publicly available datasets analyzed during the current study are: #GSE4581-Arkansas myeloma database; #GSE6477- Mayo Clinic myeloma database using the Affymetrix Human Genome U133A Array, #IA15-MM Research Foundation database and #3-Agnelli Myeloma dataset on Oncomine platform.

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
