# Peer review of "GFI1-Dependent Repression of SGPP1 Increases Multiple Myeloma Cell Survival"

_cancers, 2022, doi:10.3390/cancers14030772_

Round 1
Reviewer 1 Report
In this manuscript Petrusca et al report that GFI1 directly binds to and represses SGPP1 gene transcription, thus increasing MM cell survival in a p53 independent manner. GFI1 occupancy on two distinct region of SGPP1 is confirmed by ChIP-qPCR. GFI1-mediated SGPP1 repression significantly affects intracellular S1P levels. The authors further investigate the mechanism of S1P-dependent survival of MM cells, and report intracellular S1P levels modulate c-Myc protein stability in a PP2A-dependent manner. c-Myc is a multifunctional transcription factor which drives the multiple synthetic functions necessary for MM malignancy.
The work is of novelty and important to the field, which provides insights into how to modulate the levels of specific bioactive lipid components to affect cancer cell fate by targeting lipid metabolism, therefore providing new and attractive therapeutic approaches for MM. It for the first time identifies SGPP1 as a new target of GFI1, and follows with its functions validation in MM. The manuscript is well written and data are nicely presented. The methods used are solid and thorough to support the conclusion. Therefore, the reviewer agrees to accept the manuscript in present form.
The reviewer has one suggestion for the study which could be implemented in future endeavors. To further validate c-Myc is the key factor mediating the impact of GFI1 on MM cell survival, a degradation resistant mutant of c-Myc may be introduced to examine whether it abolishes the impact of GFI1 on S1P-dependent survival of MM cells.
Reviewer 2 Report
The manuscript by Petrusca et al reported on the observations centered around GFI1 increase multiple myeloma cell survival that includes 1) GFI1 dependent MM cell survival mediated by S1P biosynthesis regardless of p53 status, 2) the mechanism of GFI1 regulated S1P dependent survival of MM cells. The authors found GFI1 upregulates S1P by repressing SGPP1 expression, then a high level of S1P keeps PP2A inactive which stabilizes c-Myc protein, thus increasing multiple myeloma cell survival. Overall, despite some very intriguing observations with GFI1 and S1P, some issues need to be addressed by the authors.
Major comments:
The majority of the manuscript, such as figures 1,3-4,6, is talking about the S1P level, and a lot of conclusions were drawn by the levels of S1P, but it is not checked in different conditions/treatments. The lack of direct results greatly weakens the manuscript.
Minor comments include the following:
1) Figure 3C lacks internal control.
2) In result 3.3, paragraph 2, line 4, c-Myc protein expression in Fig 3F is not only slightly induced. Fig 3C, it can't be told since there is no internal control to compare.
3) In the right panel of Figure 6D, the position of P(S62)-c-Myc and I2PP2A switched compared to the left panel. Is it just the name switched?
4) In Figure S4, the arrow directions of SGPP1 and Sphk1 are inversed.
Reviewer 3 Report
Overall statement
Petrusca et al. present an orthographically and scientifically high-quality manuscript, which deals with the correlation of GIF1 and SGPP1 regarding the vitality and tumorigenicity of multiple myeloma cells. They identified the Sphingosine-1-Phosphate Phosphatase (SGPP1) gene as a novel direct target of GFI1 transcriptional repression in multiple myeloma cells, thus increasing intracellular sphingosine-1-phosphate levels, which stabilizes c-Myc.
Although I can understand all the experiments and their respective interpretation of the results very well, one thing does not reveal itself to me. There are two enzymes that can inactivate S1P; firstly the SGPP1, a simple phosphatase, and secondly the SGPL1, a lyase which degrades S1P irreversibly. At the beginning of the results section (page 7), the authors mentioned that SGPL1 was not influenced at the transcript level. These data have to be shown because the SGPP1 and SGPL1 are strongly correlated with each other on their expression levels. Furthermore, this transcript analysis must be confirmed on protein level, since SGPL1 is repressed or mutated in many tumor entities. After all, the SGPL1 inhibitor THI was also used in many experiments, which led to results comparable to those of the SPHK1 stimulator. Examples:
Page 11: Finally, we increased intracellular S1P by either upregulating its synthesis with a SphK1 pharmacological stimulator (K6PC5) and by reducing intracellular S1P irreversible degradation with a pharmacological inhibitor of S1P lyase (THI) in MM.1S GFI1 KD (Fig. 4D left panel) and OPM-2 GFI1 KD (Fig. 4D right panel). Both of these treatments significantly improved viability impaired by GFI1 depletion in these cells and had a smaller effect on the GFI1 replete control cells.
Page 14: In contrast, increasing intracellular S1P by specific S1P lyase pharmacological inhibition with THI (Fig. 6A schematic), dose-dependently increased c-Myc protein levels in MM.1S and OPM-2 cells (Fig. 6B).
Page 17: The induced intracellular S1P increase by THI significantly activated ERK1/2 in the presence and more so in the absence of GFI1, which might be responsible for the SphK1 reactivation and this could contribute to the observed c-Myc recovery in both GFI1 KD MM.1S and OPM-2 cell lines (Fig. S3E).
Thus, it cannot be ruled out that there is a sensitive equilibrium between these two enzymes, which should be taken into account in this study. I therefore recommend that the authors prove that the SGPL1 is not involved in GFI1 regulation. If this proof is given, I see no reason to oppose having this manuscript published in Cancers.
Minor issues:
- Page 2: “In the course of our earlier studies, we showed that GFI1 levels were increased by extracellular S1P.” In what context this earlier studies were performed? Makes it easier for the reader to follow, when a connection is drawn between GFI1 and S1P.
- Page 2: “S1P phosphatase 1 (SGPP1) is responsible for degrading S1P via salvage and recycling of SPH into long-chain ceramides [25, 26].” That is only half the story, because sphingosine-1-phosphate lyase is the only enzyme that can irreversibly degrade the second messenger S1P.
- To many abbreviation. Reduce it to the minimum
- Please write in vitro and in vivo in italics throughout the manuscript.
- CO2 instead of CO2
- Be consistent: h, hr or hrs.
- Page 3: 48h puromycin selection. With spaces between number and unit.
- Page 4 - 6: 30min; 1h; 30s. With spaces between number and unit.
Please go through the entire manuscript; occasionally there are still a few too many or too few spaces.
- Page 5: 0% a-MEM. Right?
- Page 7: This expression pattern was confirmed in other MM cell lines (H929-p53 WT; KMS-11-p53 null and RPMI-8226 p53 mut)(data not shown). Of note, neither SphK2 nor SGPL1 mRNA levels were altered by changes in GFI1 expression in these MM cells (data not shown)
Please include data in the supplements.
- Page 8: Explain the abbreviations o/e, VCP, WB and EV in Fig. 1 legend. It's contextual, but it makes it easier for non-molecular biology-savvy readers. Or explain it once at the beginning of the results section.
- Page 10: Fig. 3 legend, please correct: 4h, N=3.
- Page 13: Please increase the resolution of Fig. 5B.
Page 15: Fig. 6 legend, explain the abbreviation PP2A, WB. Correct the spaces.
Round 2
Reviewer 2 Report
The revised manuscript is acceptable for publication.
Reviewer 3 Report
The authors responded to all my questions and suggestions. Thank you for the detailed answers. Therefore, I recommend the publication of this manuscript.